# OpenReview forum: "WildChat: 1M ChatGPT Interaction Logs in the Wild"
_ICLR.cc/2024/Conference — ICLR 2024 spotlight_

### Official Review · Reviewer_fdao · 2023-10-28

**Soundness:** 2 fair
**Presentation:** 2 fair
**Contribution:** 2 fair
**Rating:** 5
**Confidence:** 2

**Summary:**

This paper introduces the WildChat dataset, which contains 570K user-ChatGPT dialogues with a total of 1.5 million interaction turns. The data is derived from anonymous user chat transcripts on HuggingFace, where the authors strategically deployed free chatbots in exchange for access to these interaction logs. The biggest advantange of the WildChat dataset is its significant volume of toxic content. This can greatly aid the community in analyzing toxic behavior and subsequently implementing robust models to detect such harmful content. However, it's crucial to rigorously review the toxic content prior to its release.

**Strengths:**

1. This paper deploys chatbots on HuggingFace and gathers conversational logs over a 6-month period to construct the WildChat dataset.

2. The dataset is "up-to-date" with September 2023 being the last entry.

**Weaknesses:**

1. The presentation requires a significant improvement. I suggest enhancing the writing for better clarity. Additionally, the tables and figures in the paper could benefit from improvements to enhance their readability, e.g., consider (1) increasing the spacing between the bars in Figure 1(a) for clearer visualization, (2) using `wrapfigure`  to prevent the large left and right blank margin in Figure 2, and (3) please find a way to distinguish between the overlapping blue and red scatter points in Figure 3.
2. The potential applications of the WildChat dataset appear to be restricted. While this paper has presented the dataset's utility for toxic content classification, the broader applications of this dataset remain unclear to me.
3. I understand the challenges associated with releasing the full dataset during a double-blind review process. However, it would have been beneficial if the paper had included a few sample data examples for evaluation. Additionally, it seems that the tutorial and documentation for the WildChat dataset are missing, which would have been valuable for a comprehensive review.

**Questions:**

1. I appreciate the authors' efforts in deploying chatbots on HuggingFace and gathering user interaction logs to develop the WildChat dataset.  But the intended application of the WildChat dataset is not clear. Beyond the toxic content classification, what kind of tasks researchers might undertake using this dataset?
2. The experimental assessment primarily utilizes Large Language Models (LLM) like GPT-3.5 and Llama-2 Chat. I wonder if the WildChat dataset would also be beneficial for smaller-scale language models?

**Details Of Ethics Concerns:**

The WildChat dataset provides 570K ChatGPT interaction logs, with a notable amount of potentially harmful elements, including violence and sexual content. Therefore, the release of the WildChat dataset should be strictly reviewed.

---

> ### Author Response · Authors · 2023-11-23
>
> Thank you for your comments.
>
> ### Re: The presentation requires a significant improvement. I suggest enhancing the writing for better clarity. Additionally, the tables and figures in the paper could benefit from improvements to enhance their readability, e.g., consider (1) increasing the spacing between the bars in Figure 1(a) for clearer visualization, (2) using wrapfigure to prevent the large left and right blank margin in Figure 2, and (3) please find a way to distinguish between the overlapping blue and red scatter points in Figure 3.
>
> Thanks for the suggestion! We have updated those figures in the revised draft.
>
> ### Re: The potential applications of the WildChat dataset appear to be restricted. While this paper has presented the dataset's utility for toxic content classification, the broader applications of this dataset remain unclear to me.
>
> We consider the major contribution of WildChat as being the first large-scale dataset of real use cases of ChatGPT, whereas existing datasets are mostly synthetic or use less powerful chatbots which limits the types of conversations users can have with them. WildChat can potentially be used for two types of applications: social science applications analyzing user behaviors and chatbot responses (e.g., what do people use ChatGPT for, where do chatbots fail, how often do users switch topics within a conversation, what kind of responses from chatbots will disengage users, etc.), and NLP applications such as handling toxic content or constructing better instruction tuning datasets.
>
> ### Re: I understand the challenges associated with releasing the full dataset during a double-blind review process. However, it would have been beneficial if the paper had included a few sample data examples for evaluation. Additionally, it seems that the tutorial and documentation for the WildChat dataset are missing, which would have been valuable for a comprehensive review.
>
> We have added examples at https://www.wildchatdataset.com/. We have observed that real-world conversations have a number of interesting characteristics:
> - Natural user prompts sometimes lack clarity. E.g., buying a car from a junkyard that hasnt ran since 1975
> - Users switch between multiple languages (code mixing). E.g., 英文论文中materials and methods的行文思路
> - Users change topics within a conversation
> - Users ask questions that involve multi-hop reasoning with specific domain knowledge. E.g., If there is no Invoice present in zuora revenue detail report then how to identify why it is not present though invoice is posted and revenue is correctly distributed?
>
> ### Re: The experimental assessment primarily utilizes Large Language Models (LLM) like GPT-3.5 and Llama-2 Chat. I wonder if the WildChat dataset would also be beneficial for smaller-scale language models?
>
> That’s a good point! We think that’s likely given that WildChat contains more natural conversations compared to synthetic datasets that are model-generated or expert-curated. We leave that for future work.

---

> ### Comment · Reviewer_fdao · 2023-12-01
> **Thank you for the rebuttal**
>
> I appreciate the time and effort the authors have invested in the rebuttal. It is commendable that several of my questions have been addressed. However, I have noticed that this paper still lacks a comprehensive tutorial and detailed documentation, which are essential components for a dataset paper. Therefore, I would like to keep my original score but lower the confidence to 2.

---

### Official Review · Reviewer_FR3g · 2023-10-31

**Soundness:** 3 good
**Presentation:** 4 excellent
**Contribution:** 3 good
**Rating:** 6
**Confidence:** 4

**Summary:**

This paper releases a well-collected multilingual and large-scale instruction tuning datasets collected from real user interactions with GPT-4 and GPT-3.5. Currently, improving quality and quantity of instruction tuning data is the most effective way to boost the performance of open-source large language models. The authors present very comprehensive on the statistics and analysis in terms of data distribution, quality, quantity, toxicity, and diversity. The dataset is also a multilingual dataset, which will help the minority researchers in non-English area to develop strong large language models.

**Strengths:**

1. As far as a I know, if the authors plan to release the dataset, the WILDCHAT will be the largest public instruction tuning dataset. Such instruction scale will definitely help the open-source community to construct better LLMs. Additionally, the WILDCHAT provides the coverage for 66 languages and it will help the researchers in minor languages a lot.

2. The authors adopt comprehensive and strong control and analysis on the ethical issues of the collected instruction, especially the toxicity issue.

**Weaknesses:**

1.I believe the evaluation of the instructionally tuned WILDLLAMA is too limited to demonstrate the effectiveness. In addition to MT-Bench, I strongly suggest that you can follow InstructEval (Chia et al., 2023) to evaluate WILDLLAMA on MMLU, DROP, Human-eval, and BBH. The performance superiority is not the thing to worry. Such benchmark results can help you better assess the coverage and diversity of the released WILDCHAT dataset.

2. I suggest that the authors should find a taxonomy for analyzing the task coverage, i.e. Flan, of the proposed WILDCHAT. The diverse coverage on different tasks might be more significant than the quantity. The t-SNE visualization on the diversity is not that intuitive as task coverage.

**Questions:**

1. If I understand the paper well, the 570k pool includes the harmful instructions with toxicity issues. How much instructions will be filtered and remained as harmless instructions for public release after your internal processing?

2. Can you provide the statistics on how much conversations are collected from GPT3.5-Turbo based services and GPT-4 based services, respectively? Can you also conduct other experiments to present whether fine-tuning a LLAMA2-7B model with only GPT-4 output instruction following samples might lead to better performance due to a better teacher model?

**Details Of Ethics Concerns:**

1. Please help check whether the proposed exchange between user interaction data and the free usage of OpenAI service is legal and harmless.

2. Please help check whether the released dataset is safe, which does not contain any personal information, any violations, or any other harmful information.

---

> ### Author Response · Authors · 2023-11-23
>
> Thank you for your comments.
>
> ### Re: In addition to MT-Bench, I strongly suggest that you can follow InstructEval (Chia et al., 2023) to evaluate WILDLLAMA on MMLU, DROP, Human-eval, and BBH.
>
> Thanks for the great suggestion. We have added the following evaluation. WildLlama-7b is better than Vicuna-13b on MMLU and HumanEval but worse on BBH and DROP.
>
> | Model            | Size | MMLU     | BBH      | DROP     | HumanEval |
> |------------------|------|----------|----------|----------|-----------|
> | Flan-T5-XXL      | 11B  | **54.5** | **43.9** | **67.2** | 0.0       |
> | Vicuna-13b       | 13B  | 49.7     | 37.1     | 32.9     | 15.2      |
> | GPT4 Alpaca Lora | 7B   | 35.6     | 30.7     | 27.5     | 15.9      |
> | WildLlama        | 7B   | 52.2     | 36.1     | 28.9     | **17.3**  |
>
> ### Re: I suggest that the authors should find a taxonomy for analyzing the task coverage, i.e. Flan, of the proposed WILDCHAT. The diverse coverage on different tasks might be more significant than the quantity. The t-SNE visualization on the diversity is not that intuitive as task coverage.
>
> We sampled 100 examples from each dataset and manually annotated their category, using the labels from MT-bench:
>
> |                              	| Writing | Roleplay | Reasoning | Math   | Coding  | Extraction | STEM	| Humanities | Other   |
> |----------------------------------|--------:|----------|-----------|--------|---------|------------|---------|------------|---------|
> | WildChat - (English Subset)  	| **35%** |  **20%** | 1%    	| 2% 	| 11% 	| 0%     	| 1%  	| 3%     	| 27% 	|
> | ShareGPT                     	| 27% 	| 4%   	| **2%**	| 1% 	| **30%** | 0%     	| 1%  	| 7%     	| 28% 	|
> | Alpaca                       	| 27% 	| 0%   	| **2%**	| **6%** | 2%  	| 1%     	| **19%** | 6%     	| 37% 	|
> | OpenAssistant - (English Subset) | 9%  	| 2%   	| **2%**	| 1% 	| 8%  	| 0%     	| 17% 	| **8%** 	| 53% 	|
> | Dolly                        	|	5%   |	0%	| 1%    	| 0% 	| 4%  	| **8%** 	| 9%  	| 6%     	| **67%** |
>
> Note that even under the same category there are different types of conversations. For example, the writing category in WilChat is usually open-ended, whereas in Alpaca is usually more close-ended such as short paraphrasing. We plan to include a larger scale annotation in the next version of our draft.
>
> ### Re: If I understand the paper well, the 570k pool includes the harmful instructions with toxicity issues. How much instructions will be filtered and remained as harmless instructions for public release after your internal processing?
>
> 112,455 conversations are flagged as toxic, and 459,938 conversations remain as harmless.
>
> ### Re: Can you provide the statistics on how much conversations are collected from GPT3.5-Turbo based services and GPT-4 based services, respectively? Can you also conduct other experiments to present whether fine-tuning a LLAMA2-7B model with only GPT-4 output instruction following samples might lead to better performance due to a better teacher model?
>
> 484,092 conversations were collected from GPT-3.5 and 88,301 were collected from GPT-4. We are still training a model using GPT-4 only data and will include its results in the next version of our draft.

---

### Official Review · Reviewer_dDLY · 2023-10-31

**Soundness:** 4 excellent
**Presentation:** 3 good
**Contribution:** 3 good
**Rating:** 6
**Confidence:** 4

**Summary:**

The authors compiled (INTHE)WILDCHAT, a corpus of 570K user-ChatGPT conversations, which consists of over 1.5 million interaction turns. They also study the diversity and toxicity of the corpus. The also show that fine-tuning over WILDCHAT outperforms the latest
Vicuna model of the same size on MT-Bench, which shows that WILDCHAT has a high utility.

**Strengths:**

1. The WILDCHAT dataset fills a critical gap in the available resources for the research community. The quantity of conversations surpasses the existing datasets (such as Alpaca, ShareGPT) by an order of magnitude.
2. WILDCHAT exhibits greater diversity than existing datasets, both linguistically and semantically.
3. Results demonstrates that the mere fine-tuning of a language model on the raw dataset surpasses the performance of leading open-source chatbots.

**Weaknesses:**

1. In comparing WILDCHAT with other datasets, the paper emphasizes that the token count of user prompts and assistant responses is significantly higher than that of other datasets. However, it is important to note that dialogue length does not necessarily reflect the overall quality of a dataset.
2. The paper incorporates lexical diversity as a component of the diversity of user prompts. However, whether lexical diversity alone is sufficient to reflect the diversity of user prompts. More specifically, if unigram entropy is used to calculate lexical diversity, does a scenario where each word in the user prompts is distinct necessarily indicate superior user prompts?
3. Could the toxic rate observed on Detoxify be potentially attributed to the selection of 0.1 as the threshold? This relatively low threshold might lead to false positives. Furthermore, how do the other four datasets perform on Detoxify?
4. Given that Llama-2 Chat has traded performance for alignment with humans through RLHF, one might expect its capabilities on STEM and Extraction (on MT-bench) to be somewhat diminished. Why, then, does WildLlama still fall short of Llama-2 Chat in these two areas?
5. In Table 8, WILDLLAMA still inferior to Llama-2 Chat.

**Questions:**

Please refer to the weaknesses.

---

> ### Author Response · Authors · 2023-11-23
>
> Thank you for your comments.
>
> ### Re: dialogue length does not necessarily reflect the overall quality of a dataset.
>
> We agree. We intended to show that the statistics of natural conversations are different from synthetic conversations that are expert-curated/model-generated, and dialogue length is one of such statistics.
>
> ### Re: The paper incorporates lexical diversity as a component of the diversity of user prompts. However, whether lexical diversity alone is sufficient to reflect the diversity of user prompts. More specifically, if unigram entropy is used to calculate lexical diversity, does a scenario where each word in the user prompts is distinct necessarily indicate superior user prompts?
>
> We consider diversity and quality two different aspects of data: if we randomly sample words then we maximize diversity but the quality would be terrible; if we repeat the same meaningful sentence many times then we can get good quality at the cost of diversity. In order to measure the diversity of data, we want to measure the entropy of the data distribution, and we used a simple unigram language model to approximate the data distribution. That said, we agree that unigram entropy alone is not equivalent to the diversity of user prompts, and we have revised the paragraph to be “Unigram Diversity” instead of “Lexical Diversity”.
>
> ### Re: Could the toxic rate observed on Detoxify be potentially attributed to the selection of 0.1 as the threshold? This relatively low threshold might lead to false positives. Furthermore, how do the other four datasets perform on Detoxify?
>
> Please refer to Table 9 in Appendix C for the toxicity differences between WildChat and other four datasets analyzed by Detoxify. When applying the same 0.1 threshold, WildChat has much higher toxicity ratios compared to other datasets. For example, 5.93% WildChat conversations were flagged to have explicit sexual content, while the highest percentage among other datasets is 0.08%.
>
> ### Re: Given that Llama-2 Chat has traded performance for alignment with humans through RLHF, one might expect its capabilities on STEM and Extraction (on MT-bench) to be somewhat diminished. Why, then, does WildLlama still fall short of Llama-2 Chat in these two areas?
>
> We do not know exactly what RLHF data Llama-2 Chat is trained on, but it could contain STEM and extraction tasks. On the other hand, WildChat contains very few STEM and extraction tasks, as shown in the below table, where we manually annotated 100 examples from each dataset into the task categories used in MT-bench:
>
> |                              	| Writing | Roleplay | Reasoning | Math   | Coding  | Extraction | STEM	| Humanities | Other   |
> |----------------------------------|--------:|----------|-----------|--------|---------|------------|---------|------------|---------|
> | WildChat - (English Subset)  	| **35%** |  **20%** | 1%    	| 2% 	| 11% 	| 0%     	| 1%  	| 3%     	| 27% 	|
> | ShareGPT                     	| 27% 	| 4%   	| **2%**	| 1% 	| **30%** | 0%     	| 1%  	| 7%     	| 28% 	|
> | Alpaca                       	| 27% 	| 0%   	| **2%**	| **6%** | 2%  	| 1%     	| **19%** | 6%     	| 37% 	|
> | OpenAssistant - (English Subset) | 9%  	| 2%   	| **2%**	| 1% 	| 8%  	| 0%     	| 17% 	| **8%** 	| 53% 	|
> | Dolly                        	|	5%   |	0%	| 1%    	| 0% 	| 4%  	| **8%** 	| 9%  	| 6%     	| **67%** |
>
> It is likely that WildLlama falls short of Llama-2 Chat on STEM and Extraction because WildChat contains fewer conversations from these two domains.
>
> ### Re: WILDLLAMA still inferior to Llama-2 Chat.
> Llama-2 Chat was trained using high-quality instruction tuning data, whereas we didn’t apply any data cleaning or filtering to train WildLlama; also, Llama-2 Chat underwent five additional RLHF training phases using PPO and rejection sampling finetuning with human preference data [1]. Yet, WildLlama is able to outperform other baseline chatbots, which shows the potential of the WildChat dataset as a starting point to construct a high-quality instruction tuning dataset.
>
> [1] https://arxiv.org/pdf/2307.09288.pdf

---

### Official Review · Reviewer_cWAh · 2023-10-31

**Soundness:** 3 good
**Presentation:** 3 good
**Contribution:** 3 good
**Rating:** 8
**Confidence:** 4

**Summary:**

This paper proposes a large-scale corpus of 570K user-ChatGPT conversations called WildChat. Compared with other user-chatbot interaction datasets, WildChat shows the most diverse user prompts and language usage and better aligns with real user distribution. The dataset also contains rich potentially toxic samples for research on AI safety. The authors fine-tune a chatbot on this dataset and show better performance on MT-Bench compared with the same size of the Vicuna model.

**Strengths:**

- The proposed dataset will be very valuable for LLM research communities. It will benefit the research on aligning LLM with real user prompt distribution, as well as for the safety of LLM.
- The dataset is collected under explicit user consent, the authors also try their best to protect user privacy well.
- They conduct extensive analyses on this dataset, including lexical diversity, language diversity, and data coverage, as well as toxicity, these analyses will be insightful for future research.

**Weaknesses:**

- The analysis of the dataset focuses on the toxicity aspect, along with some basic statistics. Adding more statistics such as query categories, domains, and so on, and comparing them to existing datasets, will more clearly present the information of the dataset.

- Although the data collection was done with the user's consent, I still worried about the potential privacy and legal risks, as well as toxic content. For this reason, I have requested an ethics review.

- The paper itself lacks methodological contributions, but I do appreciate the contribution of the dataset, which is also the main claim of the authors.

**Questions:**

The authors note that users are biased towards the IT domain, and that anonymization brings more harmful content, which can lead to inconsistencies with real scenarios. Is there any way to make some improvements?

**Details Of Ethics Concerns:**

Although the data collection was done with the user's consent, I still worried about the potential privacy and legal risks, as well as toxic content. For this reason, I have requested an ethics review.

---

> ### Author Response · Authors · 2023-11-23
>
> Thank you for your comments.
> ### Re: The analysis of the dataset focuses on the toxicity aspect, along with some basic statistics. Adding more statistics such as query categories, domains, and so on, and comparing them to existing datasets, will more clearly present the information of the dataset.
>
> We sampled 100 examples from each dataset and manually annotated their category, using the labels from MT-bench:
>
> |                              	| Writing | Roleplay | Reasoning | Math   | Coding  | Extraction | STEM	| Humanities | Other   |
> |----------------------------------|--------:|----------|-----------|--------|---------|------------|---------|------------|---------|
> | WildChat - (English Subset)  	| **35%** |  **20%** | 1%    	| 2% 	| 11% 	| 0%     	| 1%  	| 3%     	| 27% 	|
> | ShareGPT                     	| 27% 	| 4%   	| **2%**	| 1% 	| **30%** | 0%     	| 1%  	| 7%     	| 28% 	|
> | Alpaca                       	| 27% 	| 0%   	| **2%**	| **6%** | 2%  	| 1%     	| **19%** | 6%     	| 37% 	|
> | OpenAssistant - (English Subset) | 9%  	| 2%   	| **2%**	| 1% 	| 8%  	| 0%     	| 17% 	| **8%** 	| 53% 	|
> | Dolly                        	|	5%   |	0%	| 1%    	| 0% 	| 4%  	| **8%** 	| 9%  	| 6%     	| **67%** |
>
> Note that even under the same category there are different types of conversations. For example, the writing category in WilChat is usually open-ended, whereas in Alpaca is usually more close-ended such as short paraphrasing. We plan to include a larger scale annotation in the next version of our draft.
>
> Interestingly, we noticed a striking similarity between the above table and the evaluation results of models finetuned on each dataset on MT-bench tasks:
>
> |                   | Writing | Roleplay | Reasoning | Math    | Coding  | Extraction | STEM    | Humanities |
> |-------------------|---------|----------|-----------|---------|---------|------------|---------|------------|
> | WildLlama      | **8.6** | **8.2**  | 4.5       | 3       | **3.3** | 5.4        | **8.2** | **9.7**    |
> | Vicuna         | 7.9     | 7.3      | **4.9**   | **3.2** | 3.1     | **5.9**    | 7.3     | 9.6        |
> | Alpaca        | 6.7     | 5.5      | 3.5       | 1.1     | 2.4     | 4.2        | 5.2     | 7.9        |
> | OpenAssistant | 5.4     | 6.6      | 3.5       | 1.1     | 2.7     | 3.8        | 5.9     | 8.2        |
> | Dolly         | 4.4     | 5.2      | 2.5       | 1.6     | 1.1     | 2.5        | 4.6     | 4.2        |
>
> For example, WildChat contains the largest portion of Writing and Roleplay data, and WildLlama performs the best on these two categories.
>
> ### Re: Although the data collection was done with the user's consent, I still worried about the potential privacy and legal risks, as well as toxic content.
>
> To mitigate privacy concerns, we have anonymized the data before its release. To mitigate concerns about releasing the toxic content, we have implemented a gating mechanism to release the toxic content only upon manual approval based on the intended use of the data. To mitigate legal concerns, we have consulted with the legal department at our institution and obtained their consent.
>
> ### Re: The authors note that users are biased towards the IT domain, and that anonymization brings more harmful content, which can lead to inconsistencies with real scenarios. Is there any way to make some improvements?
>
> Since users don’t need an account to get access to our chatbot, we cannot link conversations to individual users or analyze their demographics. One way to make the dataset more balanced might be to resample it based on the conversation topics, which we leave for future work.

---

### Meta-Review · Area_Chair_3iCX · 2023-12-05

**Metareview:**

The reviewers appreciate the proposed dataset and comment on how useful it can be for the community, especially given its size and multi-linguality.

However they do raise some concerns such as the fact that the authors focus their analysis on toxicity and neglect to do the same for other categories. One reviewer raised concerns about the analysis itself and the conclusions drawn (e.g. for lexical diversity or the low threshold for toxicity detection). Another concer focuses on the evaluation of instruction-tuned WildLlama that would give insights in the data coverage, but the authors provide good responses to the above claims. Last, one reviewer commented on the clarity and presentation of the paper.

Needs ethics review before we can make a final decision, but otherwise I'm leaning towards accept.

**Justification For Why Not Higher Score:**

There is no methodological / algorithmic contribution in this work (since it is a dataset paper).

**Justification For Why Not Lower Score:**

Pending ethics review, this work has the potential to be very impactful as this is the largest dataset of its kind.

---

### Decision · Program_Chairs · 2024-01-16

Accept (spotlight)